

# Responsiveness of monopodal postural stability tests in recreational athletes

Mª Piedad Sánchez Martínez[1], Mariana Sánchez-Barbadora[2],
Noemi Moreno-Segura[2], Patricia Beltrá[3], Adrian Escriche-Escuder[2] and
Rodrigo Martín-San Agustín[2]

[1] Department of Phyiotherapy, Universidad de Murcia, Murcia, Spain
[2] Department of Physiotherapy, Universidad de Valencia, Valencia, Spain
[3] Faculty of Health Sciences, Universidad Europea de Valencia, Valencia, Spain

Corresponding author
Adrian Escriche-Escuder,
adrian.escriche@uv.es

## ABSTRACT

**Background:** Stabilometry, the modified Star Excursion Balance Test (mSEBT) or the Emery balance test (EBT) are reported in the literature to reflect changes after an intervention in monopodal postural stability. Even so, the responsiveness of those tests has not been evaluated after an instability training programme or analysed using multiple statistical indicators of responsiveness. The main aim of this study was to analyse the responsiveness of the stabilometry, mSEBT or EBT.
**Methods:** Thirty healthy recreational athletes performed a 4-week programme with three weekly sessions of instability training of the dominant lower limb and were evaluated using stabilometry, mSEBT, and EBT tests. Responsiveness was quantified based on internal and external responsiveness.
**Results:** EBT and all parameters in mSEBT for the dominant lower limb showed large internal responsiveness (SRM > 0.8). Furthermore, mSEBT values for the non-dominant lower limb (except anterior displacement) also experienced significant changes with an associated large internal responsiveness. None of the stabilometry platform parameters showed a significant change after the intervention. The ability of the EBT to discriminate between the dominant and non-dominant lower limb (*i.e.*, trained *vs* untrained, respectively) was generally acceptable (AUCs = 0.708). However, none of the parameters of the mSEBT test showed an acceptable AUC.
**Conclusions:** EBT showed a positive responsiveness after instability training compared to mSEBT, which only showed internal responsiveness, or stabilometry platform measures, whose none of the parameters could identify these changes.

## INTRODUCTION

Postural stability is defined as the ability to maintain the centre of mass of a body within the base of support with minimal postural sway through somatosensory information (*Pino-Ortega et al., 2020*), and is commonly assessed through static and dynamic balance (*Shumway-Cook & Woollacott, 2001*). Static balance is defined as the ability to maintain the line of gravity (vertical line from the centre of mass) of a body within the base of support (BoS) with minimal postural sway. While, dynamic balance consists of the ability

to move the centre of pressure (CoP) within the BoS and to move CoP from one BoS to another BoS (*Kusumoto et al., 2020*; *Reina et al., 2022*). These assessments are routinely used in sports and clinical settings to identify balance disorders. For instance, a poor balance in sports is associated with lower limb injuries (such as muscle injuries or ligament sprains) (*McGuine et al., 2000*; *Emery & Meeuwisse, 2010*; *Brachman et al., 2017*), while in the elderly population it is the most important factor associated with the risk of falls (*Muir et al., 2010*). Given its importance, the use of effective lower limb-injury detection tools is needed in order to reduce the injury rate, downtime, and health care costs associated with short- and long-term treatment of lower limb injuries (*Marcoux et al., 2017*).

Monopodal postural stability is a widely used test to assess static and dynamic balance; several tools with varying levels of difficulty have been proposed in order to adapt to the target population (*Horak, 1987*; *Emery et al., 2005*; *Powden, Dodds & Gabriel, 2019*). On the one hand, laboratory balance measures (*e.g.*, stabilometry or motion analysis) provide multiple objective values related to stability, but require the use of equipment that is costly, highly technical, and often not portable (*Horak, 1987*; *Fridén et al., 1989*; *Emery et al., 2005*; *Powden, Dodds & Gabriel, 2019*). On the other hand, other measurement tools have been developed for use in the clinical and sports setting, such as the three-directions modified Star Excursion Balance Test (mSEBT) or the Emery balance test (EBT), which are faster to perform and require less time (*Emery et al., 2005*; *Powden, Dodds & Gabriel, 2019*).

The mSEBT is the simplification in three directions of the initial eight-direction Star Excursion Balance Test described by *Gray (1995)*. It evaluates single-leg balance, dynamic neuromuscular control, proprioception, flexibility, core stability, ROM and strength while an individual reaches three directions (anterior, posteromedial, and posterolateral) with the non-stance leg (*Gribble, Hertel & Plisky, 2012*). The EBT was specifically designed to assess dynamic balance on an unstable surface with eyes closed in young adults and adolescents (*Emery et al., 2005*). The reliability and validity of these tests have been described in healthy adolescents and asymptomatic adults (*Emery et al., 2005*; *Shaffer et al., 2013*; *Powden, Dodds & Gabriel, 2019*).

These tests are reported in the literature to reflect changes after an intervention, but dissimilar results have been observed when these tests have been used simultaneously (*Blasco et al., 2019*). The clinimetric analysis of measurement instruments is of great importance in the clinical and sports settings since the change in a specific measurement can reflect a change in the patient's clinical situation, which is essential for evaluating the effectiveness of interventions (*de Yébenes Prous, Rodríguez Salvanés & Carmona Ortells, 2008*). The metric property that analyses this effect is responsiveness, which is defined as the ability of a tool to detect meaningful clinical changes over time (*Mokkink et al., 2010*). Even so, the responsiveness of monopodal postural stability measurements through stabilometry, mSEBT, and EBT has not been evaluated after an instability training programme or analysed using multiple statistical indicators of responsiveness. Furthermore, while studies use the dominant/non-dominant (*i.e.*, trained/untrained) lower limb comparison to detect within-subject changes in stability after an intervention (*Temporiti et al., 2023*), the external responsiveness (*i.e.*, discriminative ability) of the tests

has not been previously examined. Therefore, the main aim of this study was to analyse the responsiveness of the three monopodal postural stability tests.

## MATERIALS AND METHODS

### Study design

A single-group pretest-posttest design was used, which involved repeated monopodal postural stability assessment of the dominant and non-dominant lower limb before and after a 4-week intervention (three weekly sessions) consisting of dominant lower limb instability training. This study was conducted from April 2020 to June 2021, starting the recruitment phase in November 2020. All measurements were performed in the clinical research laboratory of the Department of Physiotherapy (University of Valencia). A physiotherapist with experience in applying the test (M.S-B) evaluated the participants. This examiner was blinded during the measurement process, not being aware of which limb had received the intervention. Before participation, participants were informed of the study procedures and their possible associated risks. All of them provided written informed consent. This study was completed following the principles outlined in the Declaration of Helsinki, and it was approved by the Human Research Ethics Committee of the Ethics Committee on Experimental Research of the University of Valencia (Comité Ético de Investigación en Humanos de la Comisión de Ética en Investigación Experimental de la Universitat de Valencia), in Spain (1271077).

### Subjects

Thirty healthy recreational athletes (21 males/nine females; mean age: 22.7 ± 2.7 years; weight: 70.13 ± 12.39 kg; height: 172.5 ± 8.1 cm; weekly physical activity: 438.0 ± 170.4 min) volunteered in this study, of which 27 completed the entire intervention and evaluations and were included in the analysis. Appendix S1 contains the flow chart of the study participants. Participants were physiotherapy students recruited by email using the University of Valencia Intranet. For inclusion, they had to be between 18 and 30 years old, have no history of lower limb injury or pain during the year preceding the study, and perform at least 90 min of physical activity per week. The established exclusion criteria were to have previously participated in any balance improvement or lower limb proprioception programme or presenting any known balance disorder, such as vertigo, or vestibular or central nervous system alterations.

### Instruments

#### Stabilometry

For the stabilometric assessment of monopodal stability, the Dinascan/IBV P600 force platform (digital signal with a sampling frequency of 1,000 Hz) was used with its software application NedSVE/IBV (Valencia, Spain). The participants were asked to place the foot of the leg to be measured on the mark on the platform, with the knee of the other leg flexed 90° and their arms alongside the body (Fig. 1A). The participants, with their eyes closed, were asked to maintain that position for 15 s, during which the platform recorded the variations in balance (*Romero-Franco et al., 2014*), and rested 30 s before the next
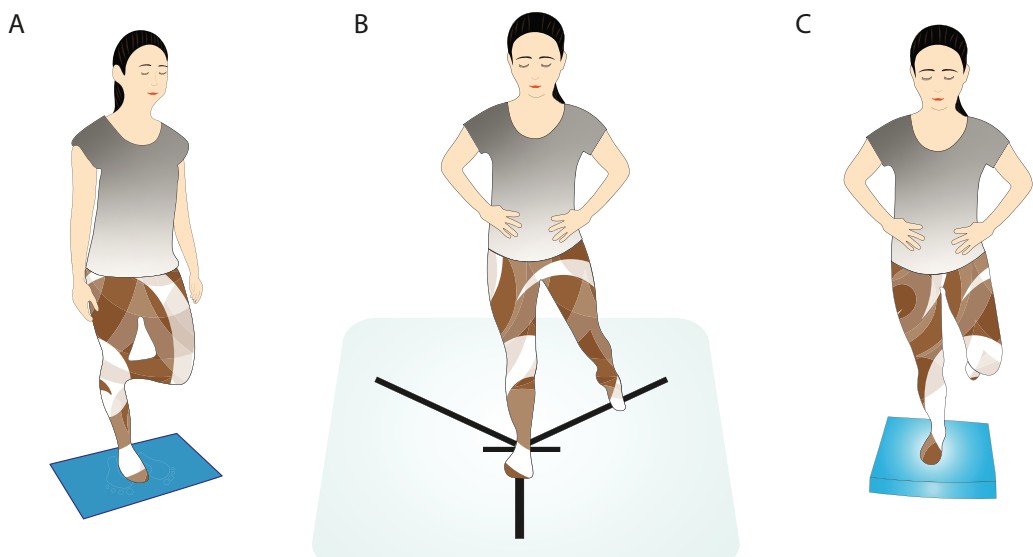

**Figure 1 Monopodal postural stability measured by (A) stabilometry, (B) modified Star Excursion Balance test, and (C) Emery balance test.**

measurement. Three measurements were taken. Subsequently, the process was repeated with the contralateral leg (*Powden et al., 2019*). The values analysed were the CoP displacement (lateral displacement and anteroposterior displacement), the swept area (mm$^2$), and the average speed (m/s). In subsequent analyses, as there is no consensus in the literature on how to process the data (*Romero-Franco et al., 2014*; *Powden et al., 2019*), stabilometry values were analysed based on four variants: the mean of the three measurements, the first measurement, the lowest, and the highest.

### mSEBT

mSEBT consists of standing on one leg while, with the contralateral leg, reaching as far as possible in three different directions (anterior, posteromedial and posterolateral) (*Plisky et al., 2006*; *Gribble, Hertel & Plisky, 2012*). Adhesive tape was placed on the floor to delimit two posterior diagonals with a 90° angle between them, with a 135° angle with respect to the anterior line (Fig. 1B). The distance covered in each attempt was normalised with the length of the leg, for which both lower limbs of each participant were measured in the supine position, taking as reference the anterior superior iliac spine and the internal malleolus of the same leg (*Gribble & Hertel, 2003*). Next, each participant was allowed to make four attempts with each leg and in each direction to practice, followed by three more attempts that were registered (*Gribble & Hertel, 2003*; *Granacher et al., 2014*). They first performed the anterior direction with their dominant leg, then the posteromedial, and finally the posterolateral. Afterwards, the same procedure was repeated with the non-dominant leg. A 15-s rest was allowed between attempts in the same position (*Granacher et al., 2014*), resting 5 min between different directions (*Gribble & Hertel, 2003*; *Granacher et al., 2014*). The values of the last three attempts were recorded to calculate the average value later.

All measurements were made barefoot and with hands placed on hips. In turn, for the anterior measurements, the stance foot was aligned at the most distal aspect of the toes, while for the backward directions, it was aligned at the most posterior aspect of the heel (Gribble, Hertel & Plisky, 2012). Attempts were not considered valid, and the movement was repeated, if the participant failed to touch the line with the mobile foot, moved the supporting foot, dropped hands from hips, lost balance at some point supporting the mobile foot, failed to maintain the start or end position for at least one second, or placed weight on the moving foot at the end of the run (Granacher et al., 2014).

### EBT

Another test used to assess the dynamic balance of a participant was the EBT, which is widely used in athletes and adolescents due to its greater complexity. Participants had to close their eyes and then stand on one leg on an Airex® Balance Pad, barefoot and with their hands placed on their hips (Emery et al., 2005; Blasco et al., 2019). The participants were asked to remain as stable as possible for a maximum time of 180 s (Hahn et al., 1999). They made three attempts with each leg and rested 15 s between them. A handheld stopwatch was used to measure the time the participant held the position. A test time of 15 s was given to the participants before starting the measurements so that they became familiar with the pad (Emery et al., 2005). The supporting leg should be slightly flexed at the knee (about 30°), and the contralateral leg should be at 45° knee flexion (Fig. 1C) (Granacher et al., 2014; Blasco et al., 2019). The recorded value was the best time obtained in the three attempts for each leg (Blasco et al., 2019). The timer was stopped when a participant dropped hands from hips, touched the ground with the contralateral leg, moved the supporting foot, moved the pad from its original position, or opened his eyes (Emery et al., 2005; Granacher et al., 2014).

### Blackboard

The instability device selected for the instability programme was the Blackboard (Blackboard Training, Innenstadt, Germany), which is a device designed to work on monopodal stability, consisting of two wooden boards joined together by tape. At its base, it has a Velcro surface on which half-cylindrical wooden bars can be freely placed. Depending on the position in which they are placed, one or other type of instability will be obtained (e.g., lateromedial or anteroposterior instability or forefoot and rearfoot only or both). The Blackboard was used in its complete instability configuration, with two bars placed in the centre of each board to create instability in both the forefoot and rearfoot (Fig. 2B).

## Procedures

Before starting the instability training programme, height was measured using a 1-millimeter sensitivity flexible tape measure, while weight and body mass index (BMI) were assessed using a standardised body composition analyser (Tanita BC 418 MA; Tanita Corp, Tokyo, Japan). In that same session, monopodal postural stability was evaluated using stabilometry, mSEBT, and EBT tests performed randomly.
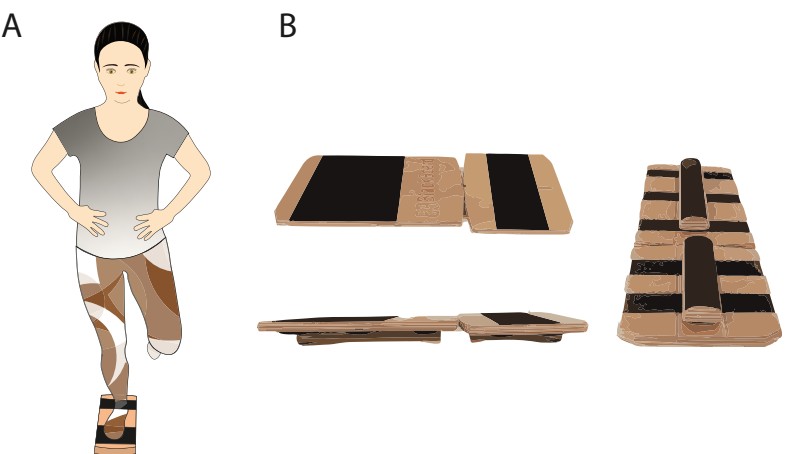

**Figure 2 (A) Stability training position using Blackboard and (B) Blackboard setup.**

A familiarisation session was then carried out in which the participants performed two to three repetitions of static single-leg support for 20 s, as needed, to become familiar with Blackboard (Fig. 2A). Next, following the same setup for the training sessions, participants performed five 40-s repetitions of training only with their dominant leg followed by 60 s of rest (*Wright, Nauman & Bosh, 2020*). The edges of the Blackboard were allowed to contact the ground and participant could slightly shift their position, but always reaching the proposed 40 s of training. Finally, a 4-week programme including three weekly sessions of instability training in order to improve the stability of the participants was performed. The duration, frequency, and dosage of the programme sessions were based on previous literature on balance training programmes (*Cain, Garceau & Linens, 2017*; *Anguish & Sandrey, 2018*; *Powden et al., 2019*), and it was carried out in a research laboratory of the Faculty of Physiotherapy of the University of Valencia.

## Statistical analysis

Baseline data were summarised as means and standard deviations (SD) for continuous variables and as absolute and relative frequencies for categorical variables. Variables were checked for normality with the Kolmogorov-Smirnov test and homogeneity of variances with Levene's test.

Responsiveness was quantified based on internal and external responsiveness. On the one hand, internal responsiveness was determined by the paired t-test and supplemented with an effect size statistic, as recommended by *Husted et al. (2000)* and similar to what was carried out by other studies (*Liang, Fossel & Larson, 1990*; *Choi et al., 2016*; *Navarro-Pujalte et al., 2019*; *Pajari et al., 2022*). For this analysis, we used the standardised response mean (SRM) as an effect size statistic, which estimates the magnitude of change that is not influenced by sample size (*Husted et al., 2000*; *Navarro-Pujalte et al., 2019*). Values of 0.20, 0.50, and 0.80 or higher have been proposed in the literature to represent small, medium, and large responsiveness, respectively (*Husted et al., 2000*).

On the other hand, external responsiveness was determined by receiver operating characteristic (ROC) curves (*Husted et al., 2000*; *Rysstad et al., 2017*; *Wan et al., 2018*; *Yee et al., 2022*). We dichotomised the values for ROC curves between the dominant and non-dominant lower limb (*i.e.*, experimental and control lower limb), assuming that the values for the dominant lower limb tests had changed after the intervention. This was done from the perspective of the responsiveness to observed change, which is quantified when scores are compared in situations where variation in the attribute is expected but not verified explicitly as having occurred (*Beaton et al., 2001*). In particular, for the circumstance of change observed before and after a treatment/intervention (usually of "known efficacy") (*Beaton et al., 2001*). We calculated the area under the ROC curve (AUC), which represents the probability of the measure correctly classifying participants. An AUC > 0.70 was used as a generic benchmark to consider its discriminant ability acceptable (*Stratford, Binkley & Riddle, 1996*). The person responsible for the statistical analysis for external responsiveness (R.M-SA) was blinded with respect to the limb in which the intervention was carried out.

An *a priori* sample size calculation was developed based on a medium effect size (d = 0.50), using an $\alpha$ value of 0.05 and a power of 0.8. The sample size was estimated at 27 subjects. Assuming losses of 10% of the sample in the follow-up measurement, an initial sample of 30 subjects was calculated as necessary.

# RESULTS

## Changes associated with instability interventions

Table 1 shows the changes associated with an instability training programme measured with three monopodal postural stability tests. The dynamic balance for the dominant lower limb, as measured with the mSEBT and EBT, showed significant time improvements and distance reached, respectively, after the interventions. For the non-dominant lower limb, a significant change was observed in the total score of the mSEBT test and in the postero-medial and postero-lateral directions. Conversely, platform measures suggested that neither limb presented significant changes in the CoP excursions after the interventions, except for the X-axis for the dominant lower limb of the first measurement recorded. Furthermore, relative changes showed the greatest improvements for EBT of the dominant leg, with a 46.2% improvement over baseline time. Appendix S2 shows individual values for all participants and tests (of the dominant lower limb).

## Internal and external responsiveness

Internal responsiveness to instability training of the three monopodal stability tests is shown in Table 2. Internal responsiveness statistics suggest that EBT and all parameters in mSEBT for the dominant lower limb showed large internal responsiveness (SRM > 0.8) among participants after instability training. Furthermore, mSEBT values for the non-dominant lower limb (except anterior displacement) also experienced significant changes with an associated large internal responsiveness. Finally, none of the stabilometry platform parameters showed a significant change in response after the intervention.

**Table 1 Differences in the dominant and non-dominant lower limb for the three monopodal stability tests after instability training.**

| | Dominant lower limb | | | Non-dominant lower limb | | |
|---|---|---|---|---|---|---|
| | Pre Mean (SD) | Post Mean (SD) | Differences Mean (95% CI) | Pre Mean (SD) | Post Mean (SD) | Differences Mean (95% CI) |
| **EBT (s)** | 11.95 (7.55) | 17.48 (9.83) | −5.52 [−8.93 to −2.12]* | 11.60 (8.22) | 11.39 (8.69) | 0.21 [−2.34 to 2.77] |
| **mSEBT** | | | | | | |
| ANT (%) | 65.0 (5.24) | 66.3 (4.83) | −1.19 [−2.23 to −0.15]* | 65.6 (5.0) | 66.4 (4.8) | −0.76 [−1.99 to 0.47] |
| PM (%) | 84.0 (12.5) | 93.6 (12.1) | −9.54 [−12.48 to −0.61]* | 85.0 (11.6) | 92.1 (11.7) | −6.60 [−9.24 to −3.96]* |
| PL (%) | 91.4 (11.0) | 97.4 (11.23) | −5.17 [−7.5 to −2.85]* | 92.1 (13.0) | 97.0 (11.1) | −4.14 [−6.45 to −1.83]* |
| Total (%) | 80.1 (8.2) | 85.8 (8.1) | −5.30 [−6.88 to −3.72]* | 81.0 (8.5) | 85.1 (7.7) | −3.83 [−5.38 to −2.3]* |
| **Stabilometry** | | | | | | |
| **Mean of 3 measurements** | | | | | | |
| Area (mm$^2$) | 420.82 (125.92) | 418.51 (143.43) | 2.31 [−46.54 to 51.17] | 412.59 (110.26) | 437.89 (117.30) | −25.29 [−68.53 to 17.93] |
| Velocity (m/s) | 0.076 (0.01) | 0.075 (0.019) | 0.001 [−0.00 to 0.00] | 0.07 (0.01) | 0.06 (0.01) | 0.003 [0.00–0.00] |
| Xmean (mm) | 41.38 (5.47) | 40.34 (5.63) | 1.04 [−1.40 to 3.48] | 40.09 (4.54) | 39.20 (5.18) | 0.88 [−0.81 to 2.58] |
| Ymean (mm) | 55.96 (11.89) | 54.60 (9.17) | 1.35 [−4.67 to 7.38] | 54.56 (9.55) | 55.21 (11.14) | −0.64 [−4.19 to 2.89] |
| **1st measure** | | | | | | |
| Area (mm$^2$) | 465.90 (205.77) | 394.74 (105.89) | 71.16 [−20.02 to 162.34] | 405.96 (125.96) | 425.59 (152.19) | −19.63 [86.38–47.12] |
| Velocity (m/s) | 0.07 (0.01) | 0.07 (0.01) | 0 [−0.00 to 0.00] | 0.07 (0.01) | 0.07 (0.01) | 0.00 [0.00–0.00] |
| Xmean (mm) | 41.81 (5.93) | 40.35 (5.88) | 1.46 [−1.75 to 4.66] | 40.58 (6.64) | 39.86 (6.29) | 0.71 [−1.74 to 3.17] |
| Ymean (mm) | 63.0 (22.41) | 52.75 (8.72) | 10.25 [0.96–19.53] | 55.24 (12.13) | 57.59 (14.68) | −2.35 [−9.66 to 4.96] |
| **Highest measure** | | | | | | |
| Area (mm$^2$) | 544.34 (212.62) | 452.44 (210.97) | 91.9 [−16.35 to 200.16] | 514.22 (149.12) | 450.88 (204.9) | 63.34 [−16.51 to 143.20] |
| Velocity (m/s) | 0.08 (0.01) | 0.07 (0.03) | 0.01 [−0.00 to 0.01] | 0.08 (0.01) | 0.07 (0.02) | 0.008 [−0.00 to 0.02] |
| Xmean (mm) | 45.98 (6.44) | 39.73 (14.67)* | 6.25 [0.87–11.63] | 43.77 (5.33) | 39.13 (14.45) | 4.64 [−0.49 to 9.77] |
| Ymean (mm) | 67.7 (20.77) | 58.44 (23.47) | 9.26 [−0.289 to 21.41] | 67.62 (13.64) | 59.47 (24.55) | 8.15 [−1.64 to 17.94] |
| **Lowest measure** | | | | | | |
| Area (mm$^2$) | 314.86 (87.52) | 282.28 (128.83) | 32.58 [−20.38 to 85.54] | 326.09 (101.82) | 271.0 (130.27) | 55.09 [3.89–106.29]* |
| Velocity (m/s) | 0.068 (0.01) | 0.061 (0.02) | 0.007 [−0.00 to 0.01] | 0.06 (0.01) | 0.05 (0.02) | 0.007 [−0.00 to 0.01] |
| Xmean (mm) | 36.84 (5.33) | 33.28 (12.44) | 3.56 [−1.44 to 8.57] | 36.24 (4.79) | 31.41 (11.97) | 4.82 [0.43–9.22]* |
| Ymean (mm) | 45.95 (9.20) | 40.77 (15.88) | 5.18 [−1.75 to 12.11] | 46.25 (8.56) | 40.91 (16.9) | 5.33 [−1.66 to 12.33] |

Notes:
  M, mean; SD, standard deviation; mSEBT, modified Star Excursion Balance Test; EBT, Emery balance test; ANT, anterior; PM, posteromedial; PL, posterolateral.
  * Statistically significant differences between pre and post measurements.

The ability of the EBT to discriminate between the dominant and non-dominant lower limb (*i.e.*, trained *vs* untrained, respectively) was generally acceptable (AUCs = 0.708) (Table 3). However, none of the parameters of the mSEBT test showed an acceptable AUC to distinguish between trained and untrained lower limbs after the intervention (AUC < 0.6). Ultimately, none of the stabilometry parameters showed acceptable AUC either.

**Table 2 Internal responsiveness statistics for the three monopodal stability tests after instability training.**

| | Dominant lower limb | | Non-dominant lower limb | |
|---|---|---|---|---|
| | Paired *t*-test (*p*) | SRM (95% CI) | Paired *t*-test (*p*) | SRM (95% CI) |
| **EBT** | 0.003 | 2.43 [1.69–3.09] | 0.864 | −0.45 [−0.98 to 0.10] |
| **mSEBT** | | | | |
| ANT | 0.026 | 1.00 [0.42–1.55] | 0.215 | 1.00 [0.42–1.55] |
| PM | 0.001 | 9.00 [7.11–10.63] | 0.001 | 7.00 [5.49–8.30] |
| PL | 0.001 | 4.17 [3.17–5.05] | 0.001 | 2.00 [1.32–2.62] |
| Total score | 0.001 | 5.00 [3.86–6.00] | 0.001 | 4.00 [3.03–4.86] |
| **Stabilometry** | | | | |
| **Mean of 3 measurements** | | | | |
| Area | 0.923 | −0.13 [−0.64 to 0.37] | 0.241 | 3.59 [2.74–4.36] |
| Velocity | 0.720 | −0.23 [−0.73 to 0.28] | 0.045 | −2.14 [−2.75 to −1.48] |
| Xmean | 0.390 | −6.41 [−7.62 to −5.01] | 0.294 | −1.39 [−1.96 to −0.78] |
| Ymean | 0.648 | 0.5 [−0.05 to 1.03] | 0.709 | 0.41 [−0.14 to 0.94] |
| **1st measure** | | | | |
| Area | 0.121 | 0.71 [0.15–1.25] | 0.551 | 0.75 [0.19–1.29] |
| Velocity | 0.966 | 0.00 [−0.52 to 0.52] | 0.295 | −0.34 [−0.87 to 0.20] |
| Xmean | 0.359 | 29.96 [23.93–35.12] | 0.556 | 2.06 [1.37–2.68] |
| Ymean | 0.032 | 0.75 [0.19–1.29] | 0.515 | 0.92 [0.35–1.47] |
| **Highest measure** | | | | |
| Area | 0.093 | 55.7 [45.13–64.83] | 0.116 | −1.14 [−1.67 to −0.58] |
| Velocity | 0.233 | −0.51 [−1.02 to 0.01] | 0.162 | −0.62 [−1.12 to −0.09] |
| Xmean | 0.024 | −0.76 [−1.27 to −0.23] | 0.075 | −0.51 [−1.02 to 0.01] |
| Ymean | 0.130 | −3.43 [−4.18 to −2.6] | 0.099 | −0.75 [−1.26 to −0.21] |
| **Lowest measure** | | | | |
| Area | 0.218 | −0.79 [−1.30 to −0.25] | 0.036 | −1.94 [−2.52 to −1.30] |
| Velocity | 0.154 | −0.55 [−1.06 to −0.03] | 0.096 | −0.64 [−1.15 to −0.11] |
| Xmean | 0.156 | −0.50 [−1.01 to 0.02] | 0.032 | −0.67 [−1.18 to −0.14] |
| Ymean | 0.137 | −0.78 [−1.29 to −0.24] | 0.130 | −0.64 [−1.15 to −0.11] |

**Note:**
SRM, standardised response mean; CI, confidence interval; mSEBT, modified Star Excursion Balance Test; EBT, Emery balance test; ANT, anterior; PM, posteromedial; PL, posterolateral.

# DISCUSSION

To our knowledge, this is the first study that analyses the responsiveness of different monopodal stability tests in healthy participants after an instability training programme. We found that only EBT showed both internal and external responsiveness, while the mSEBT showed acceptable internal responsiveness. In contrast, none of the stabilometry platform measures exhibited responsiveness.

This study presents novel findings, as it is the first study that has used multiple statistical methods to assess the internal responsiveness (paired t-test and SRM) and external responsiveness (ROC) of three measures of monopodal stability in healthy recreational athletes. This study shows that the EBT is the only monopodal stability measure that

**Table 3 External responsiveness by areas under curve (AUC) for Emery balance test and modified Star Excursion Balance Test.**

| Test | Area under curve | 95% confidence interval |
|---|---|---|
| EBT | 0.708 | [0.57–0.84] |
| mSEBT | | |
| Anterior | 0.561 | [0.40–0.71] |
| Posteromedial | 0.617 | [0.46–0.76] |
| Posterolateral | 0.557 | [0.40–0.71] |
| Total score | 0.460 | [0.31–0.62] |

Note:
    mSEBT, modified Star Excursion Balance Test; EBT, Emery balance test.

detects changes after an instability training programme, with an acceptable internal and external responsiveness. Until now, no study had analysed this psychometric ability of the EBT. However, previous studies have identified changes in stability measured using this test after an instability training programme, as reported by *Blasco et al. (2019)*. These authors found improvements in the time of the EBT (ranging between 3.3 and 6.1 s) similar to those found in our study (5.52 s) (*Blasco et al., 2019*).

Regarding the dynamic stability measured with the mSEBT, our study shows a high internal but not external responsiveness. Both the intervention and control lower limb improved for all directions, except for the anterior direction of the control side. For the intervention lower limb, all mSEBT parameters showed significant improvements. Similar results have been reported in the total score of mSEBT by *Blasco et al. (2019)*, with slightly smaller improvements (ranging between 3.2% and 4.5%) than those observed in our study (5.3% intervention lower limb). Even so, the control lower limb also exhibited similar improvements (3.8%), which, together with the lack of external responsiveness, would suggest that mSEBT is not a suitable test to monitor changes in dynamic balance using the non-dominant lower limb as control. A possible explanation is that the balance intervention on the dominant lower limb favours it going further during the mSEBT when it is not the support lower limb. Another possible mechanism is the effect of cross-education, which is defined as adaptation of an untrained limb after unilateral training of the contralateral limb (*Son & Kang, 2020*) and whose improvements appear to reflect use-dependent plasticity within the central nervous system (*i.e.*, interhemispheric communication in the brain, primarily through the *corpus* callosum) (*Lawry-Popelka, Chung & McCann, 2022*).

Another important finding of our study is that none of the stabilometry platform measures were able to detect a change in monopodal stability after the instability training programme. This is consistent with other authors who, after instability training, have found no changes in either healthy individuals (*Blasco et al., 2019*) or participants with chronic ankle instability (CAI) (*McKeon et al., 2008*). In this latter case, they concluded that CoP-based measures most likely lacked the sensitivity to detect improvements in postural control associated with a balance training programme in patients with CAI (*McKeon et al., 2008*). The fact that only the dynamic measurements showed

responsiveness compared to the measurements obtained with the stabilometric platform could be due to the fact that a healthy participant's capacity for improvement in static balance is minimal, and there is a ceiling effect for the measurements of the stabilometric platform. On the other hand, the improvement capacity for dynamic balance is possibly greater in those participants and therefore, dynamic balance-related tests detect changes.

Among the strengths, this research primarily evaluated the responsiveness of several monopodal stability tests in healthy participants. The clinical importance of this study lies in the fact that a simple and rapid dynamic test, such as the EBT, can detect changes in healthy participants after an instability training programme. This could offer a practical application in sports, where most participants are healthy. Therefore, it could be a tool used to identify whether injury prevention programmes aimed at improving monopodal stability are efficient. This study had limitations that should be considered. First, there is a limitation associated with the lack of generalisability. Thus, the sample included only healthy and young recreational athletes, so these findings cannot be extended to identify changes concerning recovery from injuries, such as knee or ankle sprains, or extrapolated to unhealthy or older populations. Even so, in view of the studies that use such tests in healthy subjects, we consider this analysis necessary, and future studies should replicate this metric platform analysis in specific populations. Secondly, the protocol used to measure stabilometry is not standardised as there is no consensus in the literature, making it difficult to compare our findings with other studies. However, we rely on the protocol proposed by *Romero-Franco et al. (2014)* to assess stabilometry measurements (*Romero-Franco et al., 2014*) while analysing stabilometry values for different variants.

## CONCLUSIONS

According to the results, a positive responsiveness of the EBT to changes in monopodal stability after instability training in healthy participants can be concluded. In contrast, mSEBT only showed internal responsiveness, and none of the stabilometry platform measures were able to identify these changes, so the stabilometry platform would not be recommended in healthy participants, as well as the mSEBT for those cases where they carry out comparisons between lower limb intra-subject.

### Funding
The authors received no funding for this work.

### Competing Interests
The authors declare that they have no competing interests.

### Author Contributions
- Mª Piedad Sánchez Martínez performed the experiments, analyzed the data, authored or reviewed drafts of the article, and approved the final draft.

- Mariana Sánchez-Barbadora conceived and designed the experiments, performed the experiments, analyzed the data, authored or reviewed drafts of the article, and approved the final draft.
- Noemi Moreno-Segura analyzed the data, authored or reviewed drafts of the article, and approved the final draft.
- Patricia Beltrá performed the experiments, analyzed the data, prepared figures and/or tables, authored or reviewed drafts of the article, and approved the final draft.
- Adrian Escriche-Escuder analyzed the data, authored or reviewed drafts of the article, and approved the final draft.
- Rodrigo Martín-San Agustín conceived and designed the experiments, performed the experiments, analyzed the data, authored or reviewed drafts of the article, and approved the final draft.

## Human Ethics

The following information was supplied relating to ethical approvals (*i.e.*, approving body and any reference numbers):

This study was approved by the Ethics Committee of the University of Valencia (Spain) (Ethical Application Ref: 1271077).

## Data Availability

The raw measurements are available in the Supplemental Files.

## Supplemental Information

Supplemental information for this article can be found online at http://dx.doi.org/10.7717/peerj.16765#supplemental-information.

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
