# Peer review of "Responsiveness of monopodal postural stability tests in recreational athletes"

_PeerJ, doi:10.7717/peerj.16765_

## Round 0.1 · original submission · Minor Revisions

The manuscript is very well written.

Minor comments:
- was there an a-priori sample size calculation?
- for the Y-balance test/ modified SEBT test - what was the testing order? which leg was tested first? which direction?
- for the Y-balance test/ modified SEBT test - The 3 trials for one direction and one stance-leg were performed once (with the 15 s pause between trials) or the stance leg was changed between trials? Was there a learning effect?
- Table 1 - there is ”*” in some rows; its meaning should be explained in legend.

·

Basic reporting

- The manuscript is very well-written in clear, professional English. The narrative flows logically.
- The literature review effectively establishes context and significance. More studies could be cited on responsiveness statistics.
- Article structure follows scientific reporting guidelines. Figures and tables clearly present results. Raw data is supplied.
- The study design and results are adequately described in a self-contained manner.

Experimental design

- This appears to be an original research study using appropriate methods to address the stated aims. - The research fills a gap around responsiveness of balance tests.
- The research question and hypotheses are clearly defined. Rationale links to current knowledge gaps.
- Methods and procedures are reported in sufficient detail to allow replication. The study design followed ethical guidelines.
- A within-subjects pre-post design was suitable to evaluate responsiveness. Blinding of assessors could have improved bias control.

Validity of the findings

- The results overall seem statistically sound, with appropriate tests used. Individual values for all participants could be shown.
- Conclusions directly relate to the research question and do not overreach the results.
- Raw data was supplied to allow scrutiny. Potential limitations around generalizability could be addressed.

Reviewer 2 ·

Basic reporting

No comment.

Experimental design

Three directions of the Star Excursion Balance Test were used in this study. However, this test is referred to as the Y-Balance Test.

The Star Excursion Balance Test (SEBT), was initially described by Gray. Several researchers used only three lines including anterior (ANT), posteromedial (PM), and posterolateral (PL) of SEBT. This “simplified” version is named in the literature as the modified SEBT. For the modified SEBT, 3 tape measures are attached to the floor.

Plisky et al used a Y or “peace sign,” incorporating the A, PM, and PL directions in the preseason screening of high school basketball players, which in turn led to the development of the YBT. It involves the individual standing on an elevated central plastic footplate 1 in (2.54 cm) off the ground and pushing a rectangular reach indicator block with the foot along a 1.5-m length of plastic tubing in each of the 3 directions.

Validity of the findings

No comment.

Additional comments

No comment.

·

Basic reporting

No comment

Experimental design

No comment

Validity of the findings

No comment

Additional comments

Review in pdf file

---

## Round 0.2 · accepted · Accept

The authors have addressed all comments.